# Photochemical Consideration in the Interactions between Blood Proteins and Layered Inorganic Materials

**DOI:** 10.3390/ijms231911367

**Published:** 2022-09-26

**Authors:** Tetsuo Yamaguchi, Hyoung-Mi Kim, Jae-Min Oh

**Affiliations:** 1Department of Energy and Materials Engineering, Dongguk University, Seoul 04620, Korea; 2Biomedical Manufacturing Technology Center, Daegyeong Division, Korea Institute of Industrial Technology (KITECH), Yeongcheon-si 38822, Korea

**Keywords:** layered double hydroxide, human serum albumin, γ-globulin, fibrinogen, biocompatibility, fluorescence quenching

## Abstract

Interactions between layered double hydroxide (LDH) nanomaterials and plasma proteins according to their particle size and surface charge were evaluated. The LDHs with different particle size (150, 350 and 2000 nm) were prepared by adjusting hydrothermal treatment and urea hydrolysis and subsequent organic coating with citrate, malite and serite was applied to control the surface charge (ζ-potential: −15, 6 and 36 mV). Adsorption isotherms and Stern–Volmer plots for fluorescence quenching indicated that the human blood plasma had weak interactions toward all the types of LDHs. The adsorption isotherms did not show significant differences in the size and surface charges, while the fluorescence quenching ratio increased with the increase in the surface charge, implying that electrostatic interaction played a major role in their interactions. The fluorescence quenching of three types of plasma proteins (human serum albumin, γ-globulin and fibrinogen) by the surface charge-controlled LDHs suggested that the proteins adsorbed on the LDHs with a single layer and additional proteins were weakly adsorbed to surround the LDHs with adsorbed proteins. It was concluded that the LDH nanomaterials are fairly compatible for blood components due to the protein corona while the electrostatic interaction can affect their interaction with the proteins.

## 1. Introduction

Development of nanotechnology expects customized healthcare by utilization of nanosized functionalized materials. Drug delivery system (DDS) has attracted attention owing to the importance of target specificity for cancer therapy and vaccination [1,2,3,4]. In order to develop DDS without side effects, biocompatible host material is highly recommended [4]. Layered double hydroxide (LDH) is known as a possible candidate of biocompatible material for anionic drugs [4,5,6,7,8]. LDH has a structure based on brucite (Mg[OH]_2_)-like layers where a part of the divalent metal cations are replaced by trivalent metals to develop positive charges in the sheets [9,10]. In order to compensate the positive charges, exchangeable anions present in the interlayer space, enabling immobilization of anionic drugs and its release through ion exchange [11,12]. Uptake of LDH by mammalian cells is possible by endocytosis and the process has been proven to not be cytotoxic [5]. It is also known that LDHs with the smaller lateral size (<50 nm) with Zn have higher cytotoxicity those with larger size (100–150 nm) and with Mg [6,13,14]. In addition to the size and the chemical composition, biocompatibility of nanoparticles is also strongly affected by the surface charge, particle morphology and solubility [15,16,17,18,19]. Most of the research on the biocompatibility of nanoparticles has been conducted in cultured cell lines [20,21,22,23,24]. Several studies have been carried out in systemic way; however, those studies focused on the pharmacokinetic behavior or pathological investigation [25,26,27,28]. Unfortunately, interactions between blood components and nanoparticles were not thoroughly examined, in spite of the possible intravenous injection of DDS nanoparticles. The situation was similar in LDH-based delivery carriers; increasing demand on development of the injectable LDH carrier was not satisfied by the research on hematocompatibility of the LDH nanoparticles [29,30,31,32,33]. According to a recent study [34], a LDH with a diameter of 2000 nm showed only 2% hemolysis after 24 h incubation, while the smaller ones did not show significant effect, suggesting high hematocompatibility of LDHs. In order to comprehend LDH–blood interactions in detail, it is highly recommended to investigate interaction between LDHs and blood plasma proteins.

Three representative proteins—human serum albumin (HSA, 54%), γ-globulin (Ig, 38%) and fibrinogen (Fb, 7%)—occupy most of the solute part in plasma. Fluorescence of tryptophan residues which dominate the protein fluorescence owing to the longest absorption wavelength and largest extinction coefficient is sensitive to conformational change in the proteins [35]. In this regard, protein fluorescence quenching assay has been widely utilized to evaluate the possible interactions of proteins toward drugs [36], nanoparticles [17,18,37,38,39] and polymers [19,40]. The accessibility of quenchers to the tryptophan residues and fluorescence properties of the tryptophans such as life-time, peak and band width, are related to locations of the tryptophan residues in the proteins [35,41]. The HSA has a single tryptophan exposed on the surface with fluorescence lifetimes of 7.8 ns and 3.3 ns due to two different molecular conformations [35,42]; Ig includes several tryptophans on the surface and inside to show three lifetimes of 0.6, 1.49 and 4.65 ns, respectively, in 10 mM phosphate buffer [43]. The fluorescence lifetime of tryptophan residues in Fb which are buried inside was estimated 2.1 ns as F-actin [41]. It has been reported that tryptophan residues with the longer fluorescence lifetime are at the periphery of the proteins, enabling higher accessibility by quenchers. All the proteins have an anionic feature at neutral pH due to the isoelectric point (pI) below 7.4 (HSA: 4.7 [44], Ig: 6.4 [44] and Fb: 5.8 [45]). Therefore, positively charged particles are expected to strongly interact with the plasma proteins compared to negative ones [32,39,46].

In this study, the biocompatibility of LDH in blood was investigated from the aspects of size, surface charges and morphologies of LDH particles. Three LDHs: small-, medium- and large-sized LDHs (designated as LDH-S, LDH-M and LDH-L, respectively) were synthesized for particle size dependence, where the size of LDH-L was not suitable for a practical carrier due to its large lateral size (~2000 nm) while LDH-L was used for experimental comparison. LDH-S was coated with three organic agents to control the surface charges (designated as LDH-S(−), LDH-S(0) and LDH-S(+), according to their surface charges). The interactions among the LDHs and the plasma proteins were investigated by both adsorption isotherms and fluorescence quenching assay. The advantages of fluorescence quenching assay over isotherm analyses will be discussed.

## 2. Results and Discussion

### 2.1. XRD Patterns

XRD patterns (Appendix A) indicated all the LDHs had a hydrotalcite phase corresponding to the Joint Committee on Powder Diffraction Standards (JCPDS) No. 38–487. The diffractograms exhibited well-developed (00l) planes with characteristic lattice peaks attributable to (012), (015), (018), (110) and (113) at 2° = 35.0°, 39.5°, 47.1°, 60.9°, and 62.2°, respectively. All the six LDHs were proven to be single-phased nanomaterials without any significant impurities. Furthermore, the coating agents for the surface charge control did not intercalate in the interlayer space of LDH-S.

### 2.2. Scanning Electron Microscopic (SEM) Images

The particle sizes and shapes of the LDHs were investigated by SEM measurement. As shown in Figure 1, a representative morphology of LDH-L seemed platelet and the other LDHs had short hexagonal column morphologies. The lateral size and thickness are summarized in Table 1. Particle size showed narrow and homogeneous distribution of the lateral sizes as illustrated in Figure 1d–f. The relatively larger lateral size (>100 nm) compared to the thickness (<100 nm) indicated anisotropy of all the LDH particles. It was noteworthy that the particle sizes and shapes of the surface charge-controlled LDHs (LDH-S(−), LDH-S(0) and LDH-S(+)) were not significantly different from the precursor, LDH-S (Figure 1 and Table 1), indicating that only the surfaces of LDH-S were thinly coated with the organic layer preserving most of physicochemical properties.

### 2.3. Dynamic Light Scattering (DLS) Analysis

In addition to the estimation of the primary particle sizes by SEM, evaluation of the hydrodynamic radii (R_H_) of the LDHs is important because most biological systems consist of an aqueous environment. The average values for R_H_ obtained from DLS analyses are summarized in Table 1 where R_H_ was slightly larger than the primary particle sizes obtained from SEM for all the LDHs. The discrepancy between the primary particle size and R_H_ was attributed to the potential agglomeration among the LDH particles. Considering the strong face-to-edge interactions among the LDH particles, the degree of the agglomeration in this research was negligible, showing that only two or three particles would come together under aqueous conditions. The R_H_ values of the charge-controlled LDHs, i.e., LDH-S(−), LDH-S(0) and LDH-S(+), were comparable with the value of the pristine LDH-S, suggesting that the hydrodynamic properties of LDH-S were not significantly altered by the surface coating.

Surface charges of nanomaterials seriously influence their biological fate as interfacial reaction is often controlled by electrostatic interaction [47]. As shown in Appendix A, LDH-S, LDH-M and LDH-L had positive zeta potentials of 35.9, 33.7 and 42.3 mV with narrow distribution at pH 7.4, respectively. The zeta potential for LDH-S(−), LDH-S(0) and LDH-S(+) was successfully controlled by the organic coating (Appendix A). A key point in the surface charge control is number of anionic and cationic centers in the coating agents. The coating with citrate for three carboxylate sites changed the surface positive charges of the pristine LDH-S to negative ones to a great degree (−15.3 mV), on the other hand, the surface charge decreased less by coating with malate for two carboxylate sites (LDH-S(0), 5.68 mV). In addition, no significant change in the surface charge was observed upon the coating with serine with one carboxylate and one ammonium site (36.5 mV) compared to that of the pristine LDH-S (35.9 mV).

### 2.4. Adsorption Isotherms

The adsorption of the proteins on the LDHs reached plateaus at equilibrium concentrations less than 0.2 g/L and the maximum adsorption amounts of the proteins were less than 0.02 g/g-LDH for HSA, 0.04 g/g-LDH for Ig and 0.08 g/g-LDH for Fb, respectively (Figure 2). According to a previous report [48], the adsorption of HSA and Ig on MgAl-LDHs with lateral sizes of 30 μm showed Langmuir-type adsorption isotherms in the concentration range of 0.5–8.0 mg-proteins/mL at 5 mg-LDH/mL. The adsorption patterns of this study were similar to the previous report. It was reported that the plasma proteins acted as surfactants to disperse LDH nanoparticles, although the plasma proteins interacted with several LDH particles to form precipitate of agglomerate at high LDH concentrations [49,50,51]. It was thought that the LDHs were surrounded by protein corona—the coverage of protein moiety on the surface of the LDHs—even at low concentrations for the proteins. The surface-covered proteins are expected to enhance colloidal stability of the LDH particles in the physiological condition, giving rise to low toxicity and prolonged circulation in a systemic level [29]. It was reported that the adsorbed amount of the proteins on LDHs largely depended on the type of the solute in the buffer condition although we did not focus on the adsorbed amount [33,48,49]. The isotherm studied explains roughly how much protein interacted with the LDH surface. However, it does not inform what kind of interactions there are in the corona layer. Furthermore, the hindrance of buffer solution in adsorption prevented quantitative analysis of the LDH–protein interactions. In this regard, photochemical consideration such as fluorescence quenching could complement the isotherm analysis in terms of the detailed LDH–protein interactions.

### 2.5. Fluorescence Spectra

Fluorescence spectra of the protein solutions (0.4 mg/mL) and the protein suspension with the LDHs with different particle sizes (0.5 mg-LDH/mL) are shown in Appendix A. The spectra were modified by removing quartz emission. In order to include the emission from the quartz cuvette in analysis, the Stern–Volmer plots for several fluorophores (Equations (4)–(7)) were used for analyses of the fluorescence quenching assay.

### 2.6. Fluorescence Quenching

Figure 3 represents the concentration-dependent fluorescence quenching ratios (Q) of the LDHs with different particle sizes and surface charges for the human blood plasma. The three LDHs with different particle sizes showed similar concentration dependent Q which reached plateaus at a ratio ~0.25 (Figure 3a). Because the three major plasma proteins—HSA, Ig and Fb—have a pH below 7.4, electrostatic interaction between the negative charge of these proteins and the positively charged LDHs is expected in a physiological condition at pH ~7.4. As shown in Appendix A, zeta potential of the blood plasma containing the three different sized LDHs was similar to the blood plasma alone (−15 mV, Appendix A), suggesting that the surfaces of the LDHs were covered by protein corona regardless of the particle size. This electrostatic interaction was clearly different in the surface charge-controlled LDHs (Figure 3b). Notably, LDH-S(+) showed an almost similar quenching pattern with its pristine (LDH-S); however, the ratios drastically decreased in LDH-S(0) (Q ~0.2) and LDH-S(−) (Q ~0.09). Because the adsorption isotherms suggested that the LDHs were surrounded by the protein corona even at low concentrations regardless of the surface charges, the strength of the interactions in the protein corona was controlled by the electrostatic interaction mainly [32].

As shown in Figure 3c,d, the Stern–Volmer plots for the fluorescence quenching of the blood plasma by the LDHs showed downward curves towards the x-axis. In order to consider the two types of fluorophores, fitting with the modified Stern–Volmer plot (Equation (6), see Experimental Section) [35] was carried out and a linear relationship was obtained (Figure 3e,f) except for LDH-S(−) and LDH-S(0). The values of *f*(1) of LDH-S, LDH-M and LDH-L were between 0.25 and 0.3 (Table 2), which suggested that about 30% of the fluorescence intensity in absence of the LDHs was attributable to the plasma protein emission. It is worth noting that K_SV_(1) decreased with an increase in the size of the LDHs (Table 2). The smaller sized LDH quenched the fluorescence of the proteins more effectively than the larger ones due to the larger surface area. In the three charge-controlled LDHs, only LDH-S(+) showed a linear relationship for the modified Stern–Volmer plot. LDH-S(−) did not provide a meaningful plot (see Appendix A) due to the small Q values. In order to comprehend the quenching behavior by LDH-S(0), fitting with the Equation (7) which was a Stern–Volmer plot including three fluorophores was performed and a reasonable fitting line was obtained (Figure 3d, Appendix A). Therefore, there were two kinds of interactions between LDH-S(0) and the plasma proteins, which were thought to be due to protein dependence or the weaker electrostatic interaction with LDH-S(0) than that with LDH-S(+).

The fluorescence quenching behavior of proteins is strongly affected by the location of the fluorescent amino acid, i.e., protein type [35,52]. In order to clarify the two kinds of the interactions observed in LDH-S(0), protein dependence on fluorescence quenching was investigated as shown in Figure 4a–c. In contrast to the adsorption isotherms, fluorescence quenching ratios (Q) of the three proteins for HSA, Ig and Fb decreased with an order of LDH-S(+) > LDH-S(0) > LDH-S(−), indicating the electrostatic interaction-controlled Q. The adsorbed amount of HSA and Ig on LDH-S(0) and LDH-S(+) at concentration of 5 mg/mL was less than 10% and about 30% of the whole protein amount (Figure 2 and Appendix A), respectively, while the quenching ratio (Q) for LDH-S(0) and LDH-S(+) were over 0.2 (HSA) and 0.3 (Ig) at the equivalent concentration (5 mg-LDH/mL). Taking into account the single layer adsorption of proteins and the similar protein sizes (HSA: ca. 8 nm and Ig: ca. 15 nm) compared to effective distance of Förster type energy transfer (ca. 10 nm), quenching by the energy transfer from the proteins to LDH-S(+) was not enough to explain the higher Q values compared to the adsorption ratios. It strongly suggested that the Q values accounted for the weakly bound protein corona in addition to the tightly bound protein on the LDHs, as concentration quenching by protein–protein interactions [51] is possible even for the weakly bound moiety.

As shown in Figure 4d–f, the Stern–Volmer plots of the three proteins by the LDHs showed downward curves towards the x-axis, similar to the plots in the plasma (Figure 3). Modified Stern–Volmer plots for LDH-S(+) had a linear relationship (Figure 5g–i), corresponding to the fluorescence quenching being explained with a single Stern–Volmer constant by ignoring the background emission from the cuvette (Figure 5a). According to life-times of the proteins, quenching constants k_q_ were estimated as summarized in Appendix A. The k_q_ obtained from the three proteins were similar, suggesting that the quenching rate less depended on the molecular environment of tryptophan in the proteins.

In the LDH-S(−) systems, meaningful modified Stern–Volmer plots were not obtained due to the low fluorescence quenching ratios, where the plot for Ig seemed to be linear; however, it did not describe the L-shaped change at a low [LDH]^−1^ region. The modified Stern–Volmer plots of HSA and Fb for LDH-S(0) showed downward curves towards the x-axis, while that of LDH-S(+) curved upwards. In order to comprehend the difference, fitting of the Stern–Volmer plots by Equation (7) in the Experimental Section was carried out (solid lines in Figure 4d–f, and Table 3). The larger K_SV_(1) than K_SV_(2) indicated that HSA and Fb can be categorized into two types; (1) adsorbed on the surface of LDH-S(0) with efficient quenching (in a green circle in Figure 5b) and (2) protein corona surrounded LDH-S(0) with adsorbed proteins with inefficient quenching (in a blue circle in Figure 5b). The same K_SV_(1) (0.820) and K_SV_(2) (0.817) for Ig strongly suggested the good fitting of the Stern–Volmer plot by Equation (5) in the Experimental Section as shown in Appendix A, where the not linear modified Stern–Volmer plot was thought to be due to the large error at low concentrations of LDH-S(0).

Stern–Volmer constants K_SV_(1) for each protein obtained in the LDH-S(+) systems (HSA: 0.888, Ig: 0.374 and Fb: 0.570) were smaller than those obtained in the blood plasma (3.46). Blood plasma includes the three plasma proteins and solutes such as vitamins and inorganic salts. As reported previously, the protein–protein interactions were affected by a combination of proteins [51,53] and buffer [48], which may affect K_SV_.

## 3. Materials and Methods

### 3.1. Materials

Chemicals such as Mg(NO_3_)_2_·6H_2_O (99%), Al(NO_3_)_3_·9H_2_O (≥98%), sodium citrate (99%), sodium malate (≥98%), serine (≥98%), NaOH (≥98%), NaHCO_3_ (99%), urea (99%) and HCl were obtained from Sigma-Aldrich LLC, St. Louis, MO, USA and used without further purification. Ca^2+^/Mg^2+^-free Dulbecco’s phosphate buffered saline (DPBS) was purchased from Thermo Fisher Scientific. The chemicals were used without further purification.

### 3.2. Preparation of LDH Nanomaterials with Different Particle Sizes and Surface Charges

LDHs with different particle sizes were prepared by hydrothermal treatment and urea hydrolysis methods. A metal solution containing Mg(NO_3_)_2_·6H_2_O and Al(NO_3_)_3_·9H_2_O (2:1 in molar ratio) was titrated with an alkaline solution (0.9 mol/L NaOH and 0.65 mol/L NaHCO_3_) to pH ~9.5. The mixture was divided to two parts and separately put into two Teflon^®^-lined stainless steel bombs. These bombs were heated at 200 °C for 24 h and for 72 h to obtain small sized (LDH-S) and medium sized LDHs (LDH-M), respectively. A large-sized LDH (LDH-L) was prepared by a combined method of urea hydrolysis and hydrothermal treatment. An aqueous solution of Mg(NO_3_)_2_·6H_2_O, Al(NO_3_)_3_·9H_2_O and urea was prepared with molar ratio of 2:1:14. The solution was heated at 90 °C for 24 h, then the obtained white suspension was stirred at 100 °C for 24 h. The final products were centrifuged (6000 rpm, 5 min) and thoroughly washed with deionized water and stored as it was.

In order to obtain LDHs with different surface charges, LDH-S was coated by organic moieties of citrate, malate and serine for negative, neutral and positive (i.e., surface charge preservation) modification, respectively [54]. Slurry of LDH-S in deionized water (10 mg/mL) was mixed with aqueous solutions containing each organic molecule (10 mg/mL) in 1:1 volume ratio. After 3 h of vigorous stirring, the solid part was separated by centrifugation and thoroughly washed with deionized water to obtain citrate-, malate- and serine- coated LDHs (designated as LDH-S(−), LDH-S(0) and LDH-S(+), respectively).

### 3.3. Characterization

Powder X-ray diffraction patterns were measured by a Bruker AXS D2 phaser diffractometer with Ni-filtered Cu-Kα radiation (λ = 1.5418 Å). The size and morphology were analyzed with scanning electron microscopy (SEM: FEI QUANTA FEG250). Size distribution was estimated with randomly selected 100 particles from the SEM images, and fitted to normal distribution using Microsoft Excel^®^ (Microsoft Corporation, Redmond, DC, USA). Zeta potentials and hydrodynamic radii of the LDHs were obtained with ELSZ-1000 (Otsuka Electronics Co., Ltd, Osaka, Japan) after properly dispersed in deionized water. The measurements were carried out with a 660 nm laser diode. The pH of the LDH suspensions was set to ~7.4 utilizing small amounts of 0.1 M NaOH and 0.1 M HCl solutions to maintain the concentration of the LDHs and the ionic strength in the suspensions for the zeta potential measurements. The hydrothermal radii were estimated by first-order self-correlation function (*G*_1_(*τ*)) and second-order self-correlation function (*G*_2_(*τ*)) with the Einstein–Stokes Equation (1)
$$G_{2}\left(\tau \right)=1+\alpha {\left(G_{1}\left(\tau \right)\right)}^{2}$$
$$G_{1}\left(\tau \right)=\exp\left(-Dq^{2}\tau \right)$$
(1)$$D=k_{B}T/3\pi \eta d$$
where, *D*: diffusion constant, *τ*: time, *q*: diffusion vector, *d*: hydrodynamic diameter, *k_B_*: Boltzmann constant, *η*: viscosity of solvent and *T*: temperature, respectively.

The zeta-potential *ζ* was estimated from Doppler shift (Equation (2)) by using a relationship with *ζ* (Equation (3))
(2)$$\Delta V=\frac{2Vnsin(\frac{\theta }{2})}{\lambda }$$
*ζ* = *4πηV*/*Eε*(3)
where, Δ*V*: Doppler shift, *n*: refractive index of solvent, *θ*: measurement angle, *V*: voltage and *ε*: dielectric constant.

### 3.4. Adsorption Isotherm

LDH suspensions were prepared at a concentration of 10 mg-LDH/mL. The suspensions were added to protein solutions of HSA, Ig and Fb at concentrations of 0.4, 0.8, 1.2, 1.6 and 2.0 mg/mL with 1:1 volume ratio. The suspensions were stirred by an orbital shaker (FINEPCR, SH30) for 18 h and were centrifuged for 5 min. Absorbance at 280 nm of the supernatants was measured by UV-vis absorption spectrometer (Shimadzu UV-1800) to measure equilibrium concentrations (*C_e_*, mg/mL). The adsorbed amount of the proteins (*q_e_*, mg/mg-LDH) were estimated from the initial concentrations (*C*_0_, mg/mL) and *C_e_* as below.
$$q_{e}=\frac{(C_{0}-C_{e})\times V}{\left[\mathrm{LDH}\right]\times V}$$
where [LDH] is the concentration of LDHs (5 mg/mL) and *V* is volume of the suspensions.

### 3.5. Plasma Protein Fluorescence Quenching

Whole blood was obtained from a healthy volunteer under the approval of Yonsei University Wonju College of Medicine (Approval No. YWMR-12-6-030). Human plasma was obtained from the whole blood by centrifugation at 3000 rpm for 5 min, and the supernatant was diluted with Ca^2+^/Mg^2+^-free DPBS by 70 times. LDH suspensions in DPBS were added to the plasma solution at a 1:1 volume ratio to obtain the concentrations of 0, 0.25, 0.5, 1.5, 2.5, 7.5 and 10 mg-LDH/mL and the mixtures were sonicated for 1 min and placed on a thermo-finemixer (FINEPCR SH2000-DX) at 36.5 °C for 30 min. After the incubation, fluorescence intensity at 340 nm of the mixtures with excitation wavelength of 280 nm was measured utilizing a luminescence spectrometer (Perkin Elmer LS55), where 280 nm is the absorption maximum of the proteins and the proteins do not have absorption at 340 nm, as shown in Appendix A. Quenching ratios were calculated as the ratio *Q* (= (*I*_0_ − *I*)/*I*_0_), where *I*_0_ and *I* stood for fluorescence intensity for negative control and the LDH treated samples, respectively.

The quenching ratio (*Q*) was analyzed by Stern–Volmer plots with multiple fluorophores (4) [55].
(4)$$\frac{I_{0}}{I}={\left[\overset{n}{\underset{i=1}{\sum }}\frac{f\left(i\right)}{1+K_{\mathrm{SV}}\left(i\right)\left[\mathrm{LDH}\right]}\right]}^{-1}$$

([LDH]: concentration of LDH (g/L), K_SV_(*i*): Stern–Volmer constant for fluorophore *i* (L/g-LDH), *f*(*i*): fraction of fluorescence from fluorophore *i* without LDH).

When *n* = 2 and K_SV_(2) = 0, Equation (2) is obtained from Equation (1)
(5)$$\frac{I_{0}}{I}={\left[\frac{f\left(1\right)}{1+K_{\mathrm{SV}}\left(1\right)\left[\mathrm{LDH}\right]}+f\left(2\right)\right]}^{-1}$$

By subtracting Equation (2) from *I*_0_, a modified Stern–Volmer Equation (6) [35] is obtained
(6)$$Q^{-1}=\frac{1}{f\left(1\right)K_{\mathrm{SV}}\left(1\right)\left[\mathrm{LDH}\right]}+\frac{1}{f\left(1\right)}$$

Moreover, when *n* = 3, K_SV_(3)= 0, Equation (7) is obtained from Equation (4)
(7)$$\frac{I_{0}}{I}={\left[\frac{f\left(1\right)}{1+K_{\mathrm{SV}}\left(1\right)\left[\mathrm{LDH}\right]}+\frac{f\left(2\right)}{1+K_{\mathrm{SV}}\left(2\right)\left[\mathrm{LDH}\right]}+f\left(3\right)\right]}^{-1}$$

According to the definition of K_SV_(*i*), quenching rate constants were estimated with the longest fluorescence lifetime of tryptophan in this report.

## 4. Conclusions

In order to comprehend the biocompatibility of layered double hydroxide (LDH) nanomaterials to whole blood, the interactions between the LDHs with blood proteins were investigated, adding to the previous research on LDH–blood cell interaction [34]. The behavior of LDH depending on particle size and surface charge was investigated by adsorption isotherms and fluorescence quenching. The small maximum adsorption of plasma proteins indicated the generally weak interactions between LDHs and proteins. However, the adsorption isotherm did not show difference in size and surface charge of LDH, as it could not only distinguish surface-bound protein and weakly interacting moieties. On the other hand, the fluorescence quenching ratio was drastically dependent on the surface charge of the LDHs. The fluorescence quenching ratio of HSA and Ig reached 0.3 and 0.4 by the positively charged LDH even though it included the emission from the cuvette as background. It suggested that the LDH adsorbed HSA and Ig with a single layer and, in addition, the LDH cores adsorbed the proteins and were surrounded by weakly-adsorbed protein corona. The analyses by Stern–Volmer plots suggested that the strength of the interaction with protein corona increased with the increase in the surface charge of the LDHs. By combining the experimental results of adsorption isotherm and fluorescence quenching, a more detailed comprehension of interactions including protein corona is possible.

## Figures and Tables

**Figure 1 ijms-23-11367-f001:**
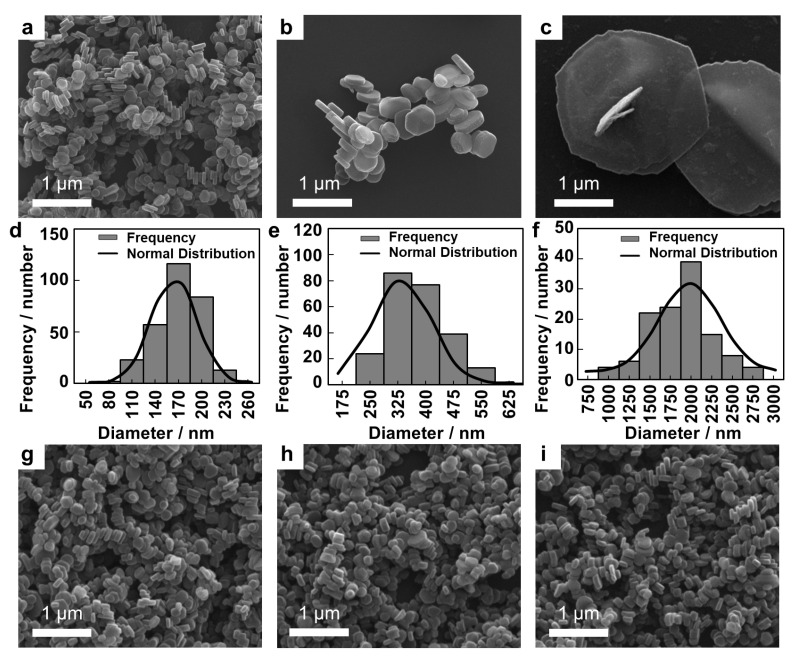
Scanning electron microscopic (SEM) images for (**a**) LDH-S, (**b**) LDH-M, (**c**) LDH-L, (**g**) LDH-S(−), (**h**) LDH-S(0) and (**i**) LDH-S(+) and size distribution of (**d**) LDH-S, (**e**) LDH-M and (**f**) LDH-L.

**Figure 2 ijms-23-11367-f002:**
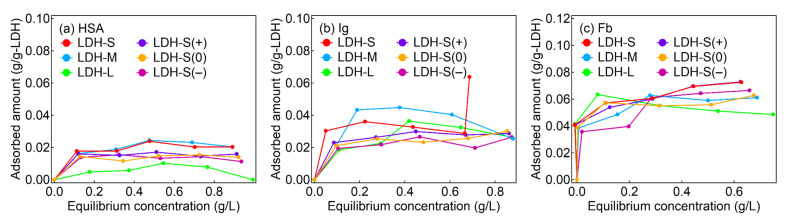
Adsorption isotherms of the proteins of (**a**) HSA, (**b**) Ig and (**c**) Fb for the LDHs.

**Figure 3 ijms-23-11367-f003:**
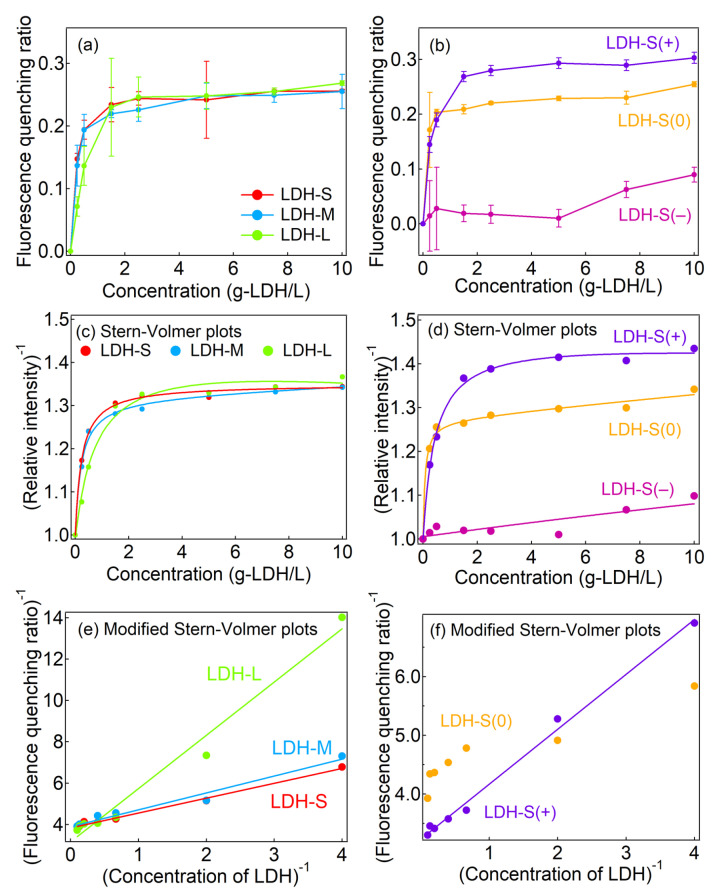
Fluorescence quenching ratios of blood plasma by LDHs from the aspects of (**a**) size and (**b**) surface charge dependence and Stern–Volmer and modified Stern–Volmer plots for fluorescence quenching for blood plasma by LDHs with different (**c**,**e**) particle sizes and (**d**,**f**) surface charges. Solid lines were obtained by fitting with Equation (7) for (**c**,**d**) and (6) for (**e**,**f**) in the Experimental Section. A modified Stern–Volmer plot for LDH-S(−) are shown in Appendix A.

**Figure 4 ijms-23-11367-f004:**
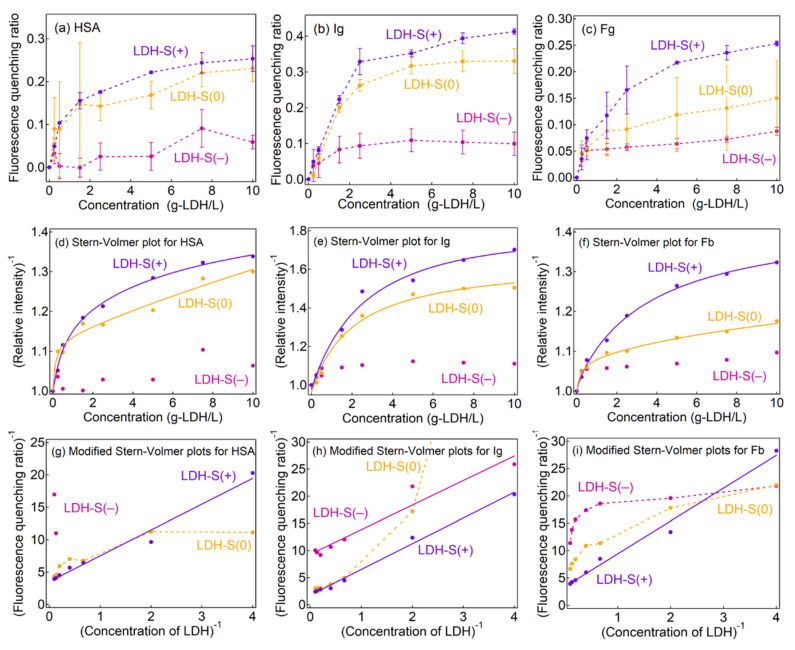
(**a**–**c**) Fluorescence quenching ratios of the three proteins, (**d**–**f**) Stern–Volmer plots and (**g**–**i**) modified Stern–Volmer plots for (left column) HSA, (middle column) Ig and (right column) Fb. Solid lines: fitting lines by (**d**–**f**) Equation (7) and (**g**–**i**) Equation (6), dotted lines: connecting the experimental data to clarify.

**Figure 5 ijms-23-11367-f005:**
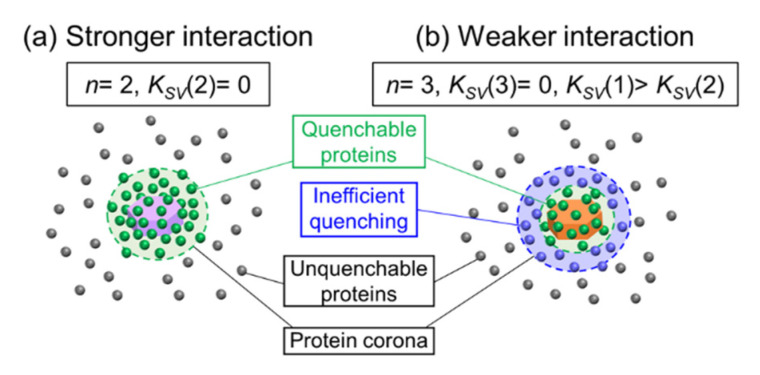
Schematic drawing of quenching of protein fluorescence by LDH with (**a**) stronger interaction with positively charged LDHs (LDH-S, LDH-M, LDH-L and LDH-S(+)) and (**b**) weaker interaction with LDH-S(0).

**Table 1 ijms-23-11367-t001:** Physicochemical parameters of the LDHs appeared in the present paper.

Sample Name	Lateral Size (nm)	Thickness (nm)	R_H_ (nm)	ζ-Potential (mV)
LDH-S	160 ± 30	60 ± 10	575	35.9
LDH-M	340 ± 70	100 ± 20	703	33.7
LDH-L	1980 ± 360	80 ± 30	2408	42.3
LDH-S(−)	160 ± 30	60 ± 10	548	−15.28
LDH-S(0)	160 ± 30	60 ± 10	595	5.68
LDH-S(+)	160 ± 30	60 ± 10	418	36.54

**Table 2 ijms-23-11367-t002:** Fitting parameters for modified Stern–Volmer plots for fluorescence quenching of blood plasma by the LDHs.

	K_SV_(1) (L(g-LDH)^−1^)	*f*(1)
LDH-S	5.32	0.261
LDH-M	4.78	0.256
LDH-L	1.22	0.317
LDH-S(−)	N.A.	N.A.
LDH-S(0)	N.A.	N.A.
LDH-S(+)	3.46	0.306

**Table 3 ijms-23-11367-t003:** Fitting parameters obtained from modified Stern–Volmer plots (6) and Stern–Volmer plots for three types of fluorophore (7) by surface charge-controlled LDHs.

Proteins	LDHs	Fitting Equation	K_SV_(1) (L(g-LDH)^−1^)	K_SV_(2) (L(g-LDH)^−1^)	*f*(1)	*f*(2)	*f*(3)
HSA	LDH-S(−)	6	N.A.		N.A.		
		7	N.A.	N.A.	N.A.	N.A.	N.A.
	LDH-S(0)	6	N.A.		N.A.		
		7	9.24	0.031	0.115	0.509	0.376
	LDH-S(+)	6	0.888		0.283		
		7	2.02	0.139	0.17	0.164	0.668
Ig	LDH-S(−)	6	2.059		0.107		
		7	N.A.	N.A.	N.A.	N.A.	N.A.
	LDH-S(0)	6	N.A.		N.A.		
		7	0.82	0.817	0.354	0.0761	0.607
	LDH-S(+)	6	0.374		0.563		
		7	0.676	−0.0108	0.51	0.117	0.394
Fb	LDH-S(−)	6	N.A.		N.A.		
		7	N.A.	N.A.	N.A.	N.A.	N.A.
	LDH-S(0)	6	N.A.		N.A.		
		7	5	0.0684	0.0739	0.179	0.747
	LDH-S(+)	6	0.57		0.292		
		7	8.17	0.368	0.0233	0.282	0.695

## Data Availability

Data are available on request due to restrictions, e.g., privacy or ethical. The data presented in this study are available on request from the corresponding author.

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
