# Peer review of "Photochemical Consideration in the Interactions between Blood Proteins and Layered Inorganic Materials"

_ijms, 2022, doi:10.3390/ijms231911367_

Round 1

Reviewer 1 Report

Review on manuscript ijms-1915930

The article is about the study of the interactions between LDH  and plasma proteins by several physico-chemical techniques. 

The article is well-written and the results are worth to be publish. 

I have some comments: 

- the purity of the applie chemical is missing (see line 283-287)

- the description of the applied techniques is poor. (e.g. DLS: laser, wavelength, Zeta: which equation was used for Zeta determination? etc. ) please complete them. line 314-316

- What was the measuring condition for Zeta experiments? Did you use inert salts?? Was the ionic strength constant? 

- For Fig. 1. the size distribution would be more informative with the SEM images. 

- For fluorescence studies did you check the absorption spectra as well? Self-absorption? Please show the absorbance spectra as well. 

Author Response

Comments by Reviewer 1:

1            The purity of the applie chemical is missing (see line 283-287)
Author Reply

              We appreciate the reviewer’s careful checking. We added the purity of the chemicals and added a sentence of “The chemicals were used without further purification.” at p. 10, line 5.

2            The description of the applied techniques is poor. (e.g. DLS: laser, wavelength, Zeta: which equation was used for Zeta determination? etc. ) please complete them. line 314-316

Author Reply

We used 600 nm laser diode for DLS and zeta-potential measurement and the particle sizes and zeta-potential were estimated by photon correlation and laser Doppler methods. In order to use photon correlation, the scattering of the laser was analyzed with first-order self-correlation function (eq. R1) and second-order self-correlation function (eq. R2) as below

         Please see attached file        (R1)

         Please see attached file          (R2)

           Please see attached file                    (R3)

Where, D is diffusion constant, τ is time, q is diffusion vector, d is hydrodynamic diameter, kB is Boltzmann constant, η is viscosity of solvent and T is temperature, respectively. .

              The Doppler shift ΔV was estimated by eq. R4 as below

       Please see attached file           (R4)

Where, n is refractive index of solvent, θ is measurement angle and V is voltage. Under electric field E, the zeta-potential ζ can be estimated from eq. R5 with using V as below.

        Please see attached file           (R5)

Where, ε is dielectric constant.

In order to reflect the reviewer’s comment, we revised sentences at p. 10, line 36 as below,

“The measurements were carried out with a 660 nm laser diode. The pH of the LDH suspensions was set to ~7.4 utilizing small amounts of 0.1 M NaOH and 0.1 M HCl solutions to maintain the concentration of the LDHs and the ionic strength in the suspensions for the zeta potential measurements. The hydrothermal radii were estimated by first-order self-correlation function (G1(τ)) and second-order self-correlation function (G2(τ)) with Einstein-Stokes equation (1)

                     Please see attached file       (1)

Where, D: diffusion constant, τ: time, q: diffusion vector, d: hydrodynamic diameter, kB: Boltzmann constant, η: viscosity of solvent and T: temperature, respectively.

The zeta-potential ζ was estimated from Doppler shift (eq. 2) with using a relationship with ζ (eq. 3)

   Please see attached file               (2)

ζ=4πηV/Eε                      (3)

Where, ΔV: Doppler shift, n: refractive index of solvent, θ: measurement angle, V: voltage and ε: dielectric constant.”

3            What was the measuring condition for Zeta experiments? Did you use inert salts?? Was the ionic strength constant?

Author Reply

              In order to simulate the pH in physiological condition (DPBS), we fixed the pH 7.4 by using small amounts of 0.1 M NaOH and 0.1 M HCl for zeta-potential measurement, because the pH affects the zeta-potential of the LDHs more than ionic strength. We believe the ionic strength is slightly difference although we did not add large amount of NaOH and HCl to maintain the concentration of the LDHs. In order to mention the point, we revised the sentence at p. 10, line 37 as below

“The pH of the LDH suspensions was set to ~7.4 utilizing small amounts of 0.1 M NaOH and 0.1 M HCl solutions to maintain the concentration of the LDHs and ionic strength in suspensions for the zeta potential measurements.”

4            For Fig. 1. the size distribution would be more informative with the SEM images.

Author Reply

              We added size distribution of LDH-S, LDH-M and LDH-L (Figure S2 in original supporting information) to Figure 1 and we revised figure caption as below. According to the addition of Figure S2 to Figure 1, the figure number was rearranged in supporting information.

Figure 1.              Scanning electron microscopic (SEM) images for (a) LDH-S, (b) LDH-M, (c) LDH-L, (g) LDH-S(−), (h) LDH-S(0) and (i) LDH-S(+) and size distribution of (d) LDH-S, (e) LDH-M and (f) LDH-L.

5            For fluorescence studies did you check the absorption spectra as well? Self-absorption? Please show the absorbance spectra as well.

Author Reply

              We appreciate and agree with the reviewer’s comment that the self-absorption can affect the emission intensity. We used 280 nm for excitation and observed 340 nm for fluorescence quenching. As we added the absorption spectra of the proteins in Fig. S6-S8 in supporting information and as below, the three types of the proteins (HSA, Ig and Fb) had absorption at 280 nm and negligibly small absorption at 340 nm. We think the effect of the self-absorption is negligible. In order to mention this point, we revised the sentence at p. 11, line 22 as below

“After the incubation, fluorescence intensity at 340 nm of the mixtures with excitation wavelength of 280 nm was measured utilizing a luminescence spectrometer (Perkin Elmer LS55), where 280 nm is the absorption maximum of the proteins and the proteins does not have absorption at 340 nm as shown in Figure S6-S8 in supporting information.”

Fig. S5   Absorption spectra of HSA at several concentrations for adsorption isotherms.

Fig. S6.  Absorption spectra of Ig at several concentrations for adsorption isotherms.

Fig. S7.  Absorption spectra of Fb at several concentrations for adsorption isotherms.

Reviewer 2 Report

The manuscript contains interesting data, it is well organized and well written.

There are some small correction to be made, see below.

1) Table 1  in order to facilitate the comparison between sample characteristics, it should be helpful to add the dimensions for the samples LDH (-) and LDH (+), even the authors mentioned in text that the 3 samples (LDH -, LDH0 and LDH + ) are in fact functionalized small LDH.

2) LDH large sample seems to be too big to be used as carrier for injectable formulations. I suggest to mention that it is studied only to compare it with the two other LDH, most suitable for parenteral administration.

3) Line 144 “It was thought that the LDHs were surrounded by protein corona which acted like surfactant even at low concentrations of the proteins” What the authors intend to say? How they presume that protein act as surfactant in formation of the corona? The author must clarify this point.

Author Response

Comments by Reviewer 2:
1.           Table 1  in order to facilitate the comparison between sample characteristics, it should be helpful to add the dimensions for the samples LDH (-) and LDH (+), even the authors mentioned in text that the 3 samples (LDH -, LDH0 and LDH + ) are in fact functionalized small LDH.

Author Reply

              We appreciate the reviewer’s suggestion. As the reviewer mentioned, Table 1 seems no data for LDH-(−) and LDH-(+). We added the data for each LDHs. Furthermore, according to the reviewer’s pointing out, it is better to specify the size information on the charge controlled LDH in the sample name. All the charge controlled LDHs started from LDH-S, and thus we revised the samples names “LDH-(−), LDH-(0) and LDH-(+)” to “LDH-S(−), LDH-S(0) and LDH-S(+)” to clarify the dimension information.

  1. LDH large sample seems to be too big to be used as carrier for injectable formulations. I suggest to mention that it is studied only to compare it with the two other LDH, most suitable for parenteral administration.

Author Reply

              We appreciate the reviewer’s comment and agree with the comment that the LDH-L is too huge to inject. In order to mention this point, we revised sentences at p. 2, line 29 as below,

“Three LDHs; small, medium and large sized ones (designated as LDH-S, LDH-M and LDH-L, respectively) were synthesized for particle size dependence, where the size of LDH-L was not suitable for a practical carrier due to its large lateral size (~2000 nm) while LDH-L was used for experimental comparison. and LDH-S was coated with three organic agents to control the surface charges (designated as LDH-(−), LDH-(0) and LDH-(+), ac-cording to their surface charges).”

  1. Line 144 “It was thought that the LDHs were surrounded by protein corona which acted like surfactant even at low concentrations of the proteins” What the authors intend to say? How they presume that protein act as surfactant in formation of the corona? The author must clarify this point.

Author Reply

              We appreciate the reviewer’s comment. In fact, the wording “surfactant” is ambiguous to describe protein corona. We would like to mention that the surface of LDHs was covered with protein and the particle was stabilized in the physiological colloids to show low toxicity. The coverage of protein in a certain degree was well described in the adsorption isotherms reached to plateau at low concentrations less than 0.2 g/L. In order to remove the “surfactant”, we revised the sentence at p. 4, line 33 from

“It was thought that the LDHs were surrounded by protein corona which acted like surfactant even at low concentrations of the proteins, hopefully giving rise to low toxicity and prolonged circulation in a systemic level [29]”

To

“It was thought that the LDHs were surrounded by protein corona, - the coverage of protein moiety on the surface of the LDHs - even at low concentrations of the proteins. The surface covered proteins are expected to enhance colloidal stability of the LDH particles in the physiological condition giving rise to low toxicity and prolonged circulation in a systemic level [29]”

Round 2

Reviewer 1 Report

I accept the corrected version 

Author Response

Comments by Reviewer 1:

1            I don't feel qualified to judge about the English language and style
Author Reply

              We revised our English again. We shows some example of the revision below.

Page 1. Abstract

“Interactions between layered double hydroxide (LDH) nanomaterials and plasma proteins according to their particle size and surface charge were evaluated. The LDHs with different particle sizes (150, 350 and 2000 nm) were prepared by adjusting hydrothermal treatment and urea hydrolysis and subsequent organic coatings with citrate, malite and serite was were applied to control the surface charge (ζ-potential: −15, 6 and 36 mV).”

Page 1, line 12,

“It is also known that LDHs with the smaller lateral sizes (< 50 nm) with Zn had higher cytotoxicity than ones with larger sizes (100-150 nm) and with Mg [6,13,14].”

Page 2, line 45

“Furthermore, the coating agents for the surface charge control did was not intercalated in the interlayer space of LDH-S.”

Page. 4, line 13

“The zeta potential for LDH-S(−), LDH-S(0) and LDH-S(+) was successfully controlled by the organic coatings (Figure S2b).”

Page 5, line 2

“Fluorescence spectra of the protein solutions (0.4 mg/mL) and the protein suspensions with the LDHs with different particle sizes (0.5 mg-LDH/mL) are shown in Figure S9.”

Page 7, line 6

“Taking into account the single layer adsorption of the proteins and the similar protein sizes (HSA: ca. 8 nm and Ig: ca. 15 nm) to effective distance…”

Page 8, line 10

“The same KSV(1) (0.820) and KSV(2) (0.817) for Ig strongly suggested the good fitting of the Stern-Volmer plot by equation (2) in experimental section as shown in Figure S12 in supporting information, where the not linear modified Stern-Volmer plot was thought to be due to the large error at the low concentrations of LDH-S(0).”

And references

“27.      Jo, M.R.; Yu, J.; Kim, H.J.; Song, J.H.; Kim, K.M.; Oh, J.M.; Choi, S.J. Titanium Dioxide Nanoparticle-Biomolecule Interactions Influence Oral Absorption. Nanomaterials 2016, 6, 225, doi:10.3390/nano6120225.”

“46.       Chen, L.; Mccrate, J.M.; Lee, J.C.M.; Li, H. The Role of Surface Charge on the Uptake and Biocompatibility of Hydroxyapatite Nanoparticles with Osteoblast Cells. Nanotechnology 2011, 22, 105708, doi:10.1088/0957-4484/22/10/105708.”
